# Theoretical insights from upscaling Michaelis-Menten microbial dynamics in biogeochemical models: a dimensionless approach

Chris H. Wilson[1], Stefan Gerber[2]

[1]Agronomy Department, University of Florida, Gainesville, FL, USA 32611
[2]Soil and Water Sciences Department, University of Florida, Gainesville, FL, USA 32611

*Correspondence to*: Chris H. Wilson (*chwilson@ufl.edu*)

**Abstract.** Leading an effective response to the accelerating crisis of anthropogenic climate
change will require improved understanding of global carbon cycling. A critical source of
uncertainty in Earth Systems Models (ESMs) is the role of microbes in mediating both the
formation and decomposition of soil organic matter, and hence in determining patterns of $CO_2$
efflux. Traditionally, ESMs model carbon turnover as a first order process impacted primarily by
abiotic factors, whereas contemporary biogeochemical models often explicitly represent the
microbial biomass and enzyme pools as the active agents of decomposition. However, the
combination of non-linear microbial kinetics and ecological heterogeneity across space and time
guarantees that upscaled dynamics will violate mean-field assumptions via Jensen's Inequality.
Violations of mean-field assumptions mean that parameter estimates from models fit to upscaled
data (e.g., eddy covariance towers) are likely systematically biased. Likewise, predictions of $CO_2$
efflux from models conditioned on mean-field values will also be biased. Here we present a
generic mathematical analysis of upscaling Michaelis-Menten kinetics under heterogeneity and
provide solutions in dimensionless form. We illustrate how our dimensionless form facilitates
qualitative insight into the significance of this scale transition and argue that it will facilitate
cross site intercomparisons of flux data. We also identify the critical terms that need to be
constrained in order to unbias parameter estimates.
**1 Introduction**
The current crisis of anthropogenic climate change is expected to accelerate during the 21st
century. Despite considerable effort to better constrain global biogeochemical models,
considerable uncertainty remains about how best to represent emerging mechanistic
understanding of soil element cycling into process-based models (Wieder et al., 2015; Todd-
Brown et al., 2018). This is a critical gap in knowledge because variations among models predict
hugely varying responses to global change drivers such as temperature, soil moisture, and $CO_2$
enrichment. For example, a traditional first-order linear model forecasts no change or even slight
enhancement of soil organic carbon (SOC) pools by 2100 whereas one microbial-explicit model
forecasts a loss of ~70Pg of carbon (C), depending on whether microbial physiology acclimates
to higher temperatures (Wieder et al., 2013). In general, our understanding of how carbon (and
other elements) cycles in soil is undergoing significant revision toward a more microbial-centric
paradigm. In contrast to traditional first-order linear models (e.g. CENTURY, Parton et al.,
1987), microbial explicit models feature non-linear dynamics in which microbial biomass (or,
similarly, microbially-driven enzyme pools) are responsible for decomposition, in addition to
providing substrate for synthesis of potentially long-term SOC (Blankinship and Schimel, 2018;
Blankinship et al., 2018). While indisputably a better representation of our scientific knowledge,
non-linear microbial models face several well-known challenges, including less analytical
tractability, greater computational challenges, and uncertainty about structural formulation and
dynamics (Georgiou et al., 2017; Sihi et al., 2016; Wang et al., 2014). However, one critical
consequence of non-linear microbial models that is only recently gaining attention is their
implications for addressing the upscaling challenge.
While the fields of population and community ecology have long confronted the challenges
posed by non-linearity and heterogeneity in spatiotemporal scaling of ecological dynamics
(Chesson, 2009; Levin, 1992), ecosystem ecology and biogeochemistry have tended to approach
the challenge of scale either by 1) utilizing mean-field assumptions, or 2) addressing the
challenge of scaling via grid-based computational/numeric methods. While there is nothing
wrong inherently with either approach, they unfortunately cannot yield theoretical insight into
the consequences of non-linearity and heterogeneity for scaling. Briefly, the combination of non-
linearity and heterogeneity means that aggregated behavior differs systematically from mean-
field predictions, a special case of Jensen's Inequality. In mathematical notation:
$$E[f(x)] \neq f(E[x]) \tag{1}$$

Although Jensen's Inequality is well-known from basic probability theory (Ross, 2002) it's
implications for ecological dynamics under heterogeneity were not well-appreciated until the
pioneering work of Peter Chesson in the 1990s (Chesson, 1998). In the case of carbon cycle
science, there are a few immediate and critical applications. For instance, most trace gas
emission processes are well-known to be non-linear functions of underlying drivers such as
temperature and soil moisture. For example, ecosystem respiration is an exponential function of
temperature (usually expressed in $Q_{10}$), and a unimodal function of soil moisture. Thus, when
matching observations of $CO_2$ efflux ("$F$") to ecosystems, variations in soil temperature and
moisture could imply that $F$ differs systematically from a mean-field prediction. Likewise,
variations in biotic interactions between microbes likely play a key role in biogeochemical
cycling (Buchkowski et al. 2017). In addition to missing critical analytical insight, not
accounting for this behavior might have severe consequences for inverse modeling and
estimation of the parameters governing process-based models (Bradford et al., 2021). Moreover,
a significant advance in recent research has focused not only on microbial-explicit formulations,
but the role of microbe-substrate colocation in the complex and heterogeneous soil environment
in both the synthesis and decomposition of organic matter (Schimel and Schaeffer 2012,
Lehmann et al. 2020). This spatial colocation itself has very important implications for scale
transitions in soil systems, and thus requires specific theoretical attention from this perspective.
Overall, the basic consequences of Jensen's Inequality for estimation of trace gas emission ($CH_4$
and $N_2O$) were first discussed by Van Oijen et al. (2017), but have not been picked up on
elsewhere, until the present work and by Chakrawal et al. (2020)
Chakrawal et al. (2020) provide a detailed and compelling first-pass application of scale
transition theory to biogeochemical modeling. Our contribution here complements their laudable
effort by providing a more generic mathematical analysis of the scale transition, equally
applicable to both forward and reverse Michaelis-Menten microbial kinetics. As in Chakrawal et
al. (2020), we address the consequences of heterogeneity in both substrate/microbes
("biochemical heterogeneity") as well as in the kinetic parameters ("ecological heterogeneity").
However, we diverge from their approach in that, rather than explore detailed simulation models,
we derive a completely non-dimensionalized expression for aggregating non-linear microbial
kinetics over both types of heterogeneity simultaneously. We illustrate the clarity this brings in
several special cases of our full analysis. Altogether, our approach provides new insight into the
properties of the scale transition and enables clear conclusions to be drawn across systems in
terms of the role of spatial variances and covariances in shaping ecosystem carbon efflux. Our
work provides a simplified, yet systematic framework around which to base subsequent
empirical and simulation-based studies.

## 2. Carbon Efflux and the Scale Transition

A variety of microbial-explicit process-based models have been proposed in the literature, starting with the classic enzyme pool model of Schimel and Weintraub (2003). In order to elucidate universal properties of the scale transition, we focus here on the $CO_2$ efflux following decomposition of a single substrate by a single microbial pool obeying Michaelis-Menten (MM) dynamics:

$$F = -f(C, MB, \theta) \tag{2}$$

where $F$ is the $CO_2$ flux, C is the carbon substrate, MB the live microbial biomass, $\theta$ is a vector of parameters, specifically $V_{max}$ (the maximum reaction rate given saturation of either C, in forward MM, or microbial biomass (MB), in reverse MM), $k_h$ (the half-saturation constant), and carbon-use efficiency, ε.

Our specific model for $F$ is:

$$F = (1 - \varepsilon) \times C \times \frac{V_{max}MB}{k_h + MB} \tag{3}$$

Following the terminology of Chesson (1998,2012), the above is our "patch" model and our goal is to understand how spatial variances and covariances impact the integrated flux, which represents the spatial expectation or E[$F$] (hereafter denoted $\overline{F}$), which represents

$$\overline{F} = \overline{-C \times (1 - \varepsilon) \times \frac{V_{max}MB}{k_h + MB}} \tag{4}$$

Where the bar over the expression represents the mean. The incorrect approach to solving for E[$F$] is to simply plug-in the mean-field solution:

$$\overline{F} = \overline{-C} \times \overline{(1 - \varepsilon)} \times \frac{\overline{V_{max}}\ \overline{MB}}{k_h + \overline{MB}} \tag{5}$$

Analytically, an exact solution would require specification of a joint distribution for C, MB and parameters, $\pi(MB, C, \theta)$, and solution of the convolution integral:

$$\iiint -f(MB, C, \theta)\pi(MB, C, \theta)dMBdCd\theta \tag{6}$$

However, following Chesson (2012) and Chakrawal et al. (2019) we are free to approximate the solution for arbitrary distributions using a Taylor Series approximation expanded to the 2nd moment. Specifically, we take the expectation over a multivariable Taylor Series expansion,

centered around the mean-field values of all parameters $\theta$ (for simplicitiy, the variables $MB$ and
$C$ are included in the parameter vector $\theta$):

$$\overline{F} \approx \mathrm{E}\left[f(\overline{\theta}) + \frac{1}{2}\theta_{\theta-\overline{\theta}}^T H_{\overline{\theta}}[f(\theta)]\theta_{\theta-\overline{\theta}}\right] \tag{7}$$

where $H[f(\theta)]$ represents the Hessian matrix of the function that determines the $CO_2$ efflux $F$
(in this case Michaelis-Menten), $\theta_{\theta-\overline{\theta}}$ represents the deviation from the mean at each instance
and for each of the parameters. It can easily be seen that $\theta_{\theta-\overline{\theta}}^T \times \theta_{\theta-\overline{\theta}}$ is the variance-covariance
matrix, and that the first moment of the Taylor expansion cancels because the first derivative of
$\theta_{\theta-\overline{\theta}}$ is zero.

**2.1 Non-dimensionalization**

Expanding equation 7 out, we have 5 terms involving the variances of $C$, $MB$, $1-\varepsilon$, $V_{max}$, and
$k_h$, and 10 terms involving covariances among the parameters. We can redistribute the
expectation operator over this approximation to see that we are dealing with the contributions
from the variance-covariance terms, weighted by the second partial derivatives evaluated at the
mean for each parameter. However, the resulting expression **does not readily yield insight into**
**the impact of scale transition upon the dynamics**, since second partial derivatives and cross
partial derivatives do not have easy intuition. Moreover, variances and covariances depend
arbitrarily upon the scale of units and measurements involved, hindering both intuition and cross-
site comparisons. Therefore, we non-dimensionalize equation 7 for $\overline{F}$ as follows:
1) We define a dimensionless quantity $\lambda$ as $\frac{\overline{MB}}{\overline{k_h}}$. $\lambda$ thus represents a multiplicative factor
expressing the ratio of the mean microbial biomass over it's half-saturation value, indicating
the microbial saturation for the decomposition.
2) We divide all of the terms in 6 by their mean-field value, and represent the whole equation
as a product:
$$\overline{F} \approx f(\overline{\theta}) + f(\overline{\theta})\left((\theta-\overline{\theta})^T \frac{\frac{\partial^2 f}{\partial\theta^2}_{\theta=\overline{\theta}}}{f(\overline{\theta})}(\theta-\overline{\theta})\right) = f(\overline{\theta})\left(1 + \left((\theta-\overline{\theta})^T \frac{\frac{\partial^2 f}{\partial\theta^2}_{\theta=\overline{\theta}}}{f(\overline{\theta})}(\theta-\overline{\theta})\right)\right)$$

158 (8)

3) We calculate the resulting expression for $\overline{F}$
4) We notice that $\frac{Var(\theta)}{\overline{\theta}^2}$ can be re-expressed as $(\frac{SD(\theta)}{\overline{\theta}})^2$ which in turn is the square of the
dimensionless coefficient of variation $(CV(\theta))^2$. This enables us to reformulate the
variance terms in (7).
5) Similarly, since the covariance terms can be rewritten as $COV(X,Y) = \rho_{X,Y}SD(X)SD(Y)$,
we have the following equality:
$$\frac{COV(X,Y)}{\overline{XY}} = \rho_{X,Y}CV(X)CV(Y) \tag{9}$$

Applying steps 1-5 to all the terms in the equation, we end up with a fully dimensionless
equation:

$$\overline{F} \approx f(\overline{\theta})(1 + \frac{\lambda}{(1+\lambda)^2}[\rho_{k_h,MB}CV(k_h)CV(MB) - CV(MB)^2] + \frac{1}{(1+\lambda)^2}[CV(k_h)^2 -$$

$$\rho_{k_h,MB}CV(h)CV(MB)] + \frac{1}{(1+\lambda)}[\rho_{C,MB}CV(C)CV(MB) + \rho_{V_m,MB}CV(V_m)CV(MB) +$$

$$\rho_{\epsilon,MB}CV(\epsilon)CV(MB) + \rho_{C,k_h}CV(C)CV(k_h) - \rho_{V_m,k_h}CV(V_m)CV(k_h) - \rho_{k_h,\epsilon}CV(k_h)CV(\epsilon)] +$$

$$\rho_{V_m,C}CV(V_m)CV(C) + \rho_{C,\epsilon}CV(C)CV(\epsilon) + \rho_{\epsilon,V_m}CV(\epsilon)CV(V_m)$$

$$(10)$$


Note that by symmetry, we have also solved for the case of the forward Michaelis-Menten
kinetics. This can be expressed simply by interchanging $C$ and $MB$, and by correspondingly
altering $\lambda$ to represent the ratio of substrate availability over half-saturation.

## 3. Discussion

Having fully non-dimensionalized equation 7, we are in a much better position to gain analytical
insight into the scale transition. To begin, we note the pivotal role played by the quantity $\lambda$
throughout this equation. $\lambda$ scales the contributions of the parameter variation and correlation
terms to the deviation from mean field behavior according to the ratios $\frac{\lambda}{(1+\lambda)^2}$, $\frac{1}{(1+\lambda)^2}$, and $\frac{1}{1+\lambda}$. All
of the parameter variance terms (which have become $CV(\theta)^2$ upon non-dimensionalization), are
scaled by one of these three $\lambda$ ratios, alongside 7 out of 10 of the covariance terms. Overall, low
$\lambda$ (here $\lambda <\approx 1$) keeps all the spatial correction terms in play, while increasing $\lambda$ tends to
simplify matters. As noted by others (Sihi et al., 2016; Buchkowski et al., 2017), as $MB \rightarrow \infty$
(equivalent to MB $>> k_h$ or $\lambda \rightarrow \infty$), reverse Michaelis-Menten kinetics converge to first order,
leaving:

$$\overline{F} = -C \times (1 - \epsilon) \times V_{max} \qquad (11)$$

Accordingly, in our setup, the multiplicative factor for the scale transition correction approaches
a simplified expression, as $\lambda \rightarrow \infty$:

$$\overline{F} \rightarrow f(\overline{\theta})\left(1 + \rho_{V_m,C}CV(V_m)CV(C) + \rho_{C,\epsilon}CV(C)CV(\epsilon) + \rho_{\epsilon,V_m}CV(\epsilon)CV(V_m)\right) \qquad (12)$$

This is quite remarkable. Despite invoking the situation where microbial biomass (and its
enzyme supply) is effectively infinite - thus linearizing the underlying patch models - we cannot
eliminate the possibility of a potentially substantial deviation from mean-field when scaling
decomposition kinetics. We note that in this resulting expression, we have reduced the situation
to a set of three critical correlations involving two microbial physiological parameters ($\epsilon$, and
$V_m$), and substrate availability ($C$). Regardless of their respective variabilities (CV terms), if
these correlations are close to zero, then the whole expression converges to mean field.
Returning to the situation where $\lambda$ is not large, if we ignore the correlation terms (temporarily
setting to zero), we see that there are direct contributions to the scale transition from the
variability in $MB$ and $k_h$ that may, to some extent, balance each other:

$$\overline{F} = f(\overline{\theta})(1 + \frac{1}{((1+\lambda)^2)}[CV(k_h)^2] - \frac{\lambda}{(1+\lambda)^2}[CV(MB)^2]) \tag{13}$$

Focusing on the offsetting correction terms, we can re-write as:

$$\frac{\lambda}{(1+\lambda)^2}[\frac{CV(k_h)^2}{\lambda} - CV(MB)^2] \tag{14}$$

and for the case of $\lambda = 1$, this becomes:

$$\frac{1}{4}[CV(k_h)^2 - CV(MB)^2] \tag{15}$$

Thus, variability in the factors of soil protection that impact upon $k_h$ in practice, can offset the
impact of variability in microbial biomass itself.
More generally, starting with our dimensionless equation 10 puts modelers and empiricists in a
better position to assess the quantitative significance of the scale transition correction across
systems compared to expressions with opaque second partial derivatives and cross derivatives,
and arbitrarily scaled variance terms. By re-expressing $\overline{F}$ in terms of dimensionless coefficients
of variation, correlation coefficients and $\lambda$, we can plug-in realistic values for variability in any
relevant parameter and assess the % effect on $\overline{F}$ in terms of deviation from mean field behavior.
We argue that this formulation possesses significant advantages not only in understanding how
to scale flux estimates ($\overline{F}$) *within* a site, but going forward will help facilitate intercomparison
*among* sites in terms of their scale-free variability. In particular, we explore variation in
dominant environmental drivers of inter-site variation (temperature and soil moisture) below. But
first, we analyze how the scale transition sheds new light on microbe-substrate colocation.

### 3.1 Spatial Colocation of Microbes and Substrate

To illustrate these advantages in interpretability, we first take the special case of a model where
we treat all parameters as constant (and known) except substrate and microbial biomass. This
corresponds to setting the other CV and $\rho$ terms to 0. In this case, we are isolating the impact of
the spatial colocation of substrate and decomposers. Our equation becomes:

$$\overline{F} \approx f(\overline{\theta})(1 - \frac{\lambda}{(1+\lambda)^2}CV(MB)^2 + \frac{1}{(1+\lambda)}(\rho_{C,MB}CV(C)CV(MB))) \tag{16}$$

In the case of this formulation, there is a very clear dual convergence as $\lambda$ increases:
1.   deviation from mean-field behavior declines, and
2.   first order kinetics are approached
Indeed, our equation 16 reveals the exact speed of this convergence in terms of dimensionless $\lambda$
and a balance of $CV(MB)$, $CV(C)$ and their correlation.
We illustrate the scale transition solutions to equation 16 as a function of $\lambda$ for various choices of
CV(C), CV(MB) and $\rho$ in Fig. 1:

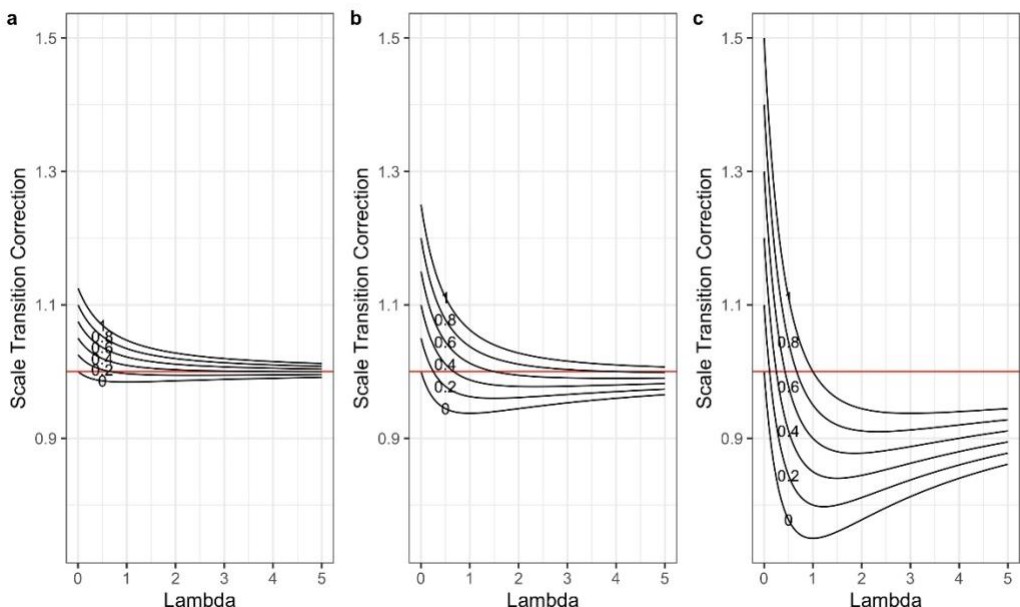


Figure 1: Scale transition correction for models given spatial colocation between microbes and substrate across a gradient of $\lambda$ values, and for a variety of correlation $\rho$ values (0-1), with CV(SOC) held constant at 0.5, a) CV(MB) = 0.25, b) CV(MB) = 0.5, and c) CV(MB) = 1.Note that in panel c) the system appears to converge on a value lower than 1. However, as $\lambda$ increases, convergence to 1 does occur, albeit slowly, as it must according to equation 16.

In the case of pure spatial colocation, with no variation in the kinetic parameters, the scale transition correction factor varies from a maximum of 1.5 to a minimum around 0.75, and in all cases indeed converges to 1 as $\lambda$ increases. The variability assumed for C and MB impacts only the scale of the correction factor, not the qualitative behavior as $\lambda$ and $\rho$ vary. One benefit of having a simplified, generic dimensionless equation of this sort is that it enables us to think in a unit-free/scale-free manner about the *plausible* range of the scale transition correction given transparent assumptions about variability and correlations.

Another benefit is that it is mathematically tractable to see how the variance and covariance terms can balance each other, and to solve for where they are equal. If we introduce a new term $\lambda_2$ representing the relationship between $CV(MB)$ and $CV(C)$ as follows $CV(MB) = \lambda_2 CV(C)$, we can re-express the the deviation of the mean-field correction from 1 as:

$$CV(MB)^2[\frac{1}{1+\lambda}\left(\rho\lambda_2 - \frac{\lambda}{(1+\lambda)}\right)] \tag{17}$$

Thus, whether the correction is positive or negative depends crucially on the product of the colocation correlation coefficient $\rho$ and the extent of variability in substrate relative to variability in microbes.

If we fix $\lambda$ to unity, as done in our Fig.1, our mean-field deviation simplifies to:

$$CV(MB)^2[\frac{1}{2}(\rho\lambda_2 - \frac{1}{2})] \tag{18}$$

In general, the scale transition correction is larger to the extent that microbial variability exceeds
substrate variability under reverse Michaelis-Menten kinetics (the opposite relation holds for
forward Michaelis-Menten by symmetry). Thus, variability in microbial biomass is not only
important by itself in driving Jensen's Inequality, but also with respect to variability in substrate
supply. **Our analysis thus highlights another route of convergence back to the mean field**
**beyond the simple increase of $\lambda$: variability in substrate increasing to match variability in**
**microbes in the presence of positive spatial colocation factor**. We also note that the
magnitude of the scale transition correction scales as the *square* of the coefficient of variation of
microbial biomass. Quadratic scaling means that at low to moderate levels of variability, the
deviation from mean field behavior is likely to be minimal, but at moderately high to high levels
of variability, severe deviations can be expected. Finally, we note that throughout, our
development of these kinetics assume proportionality to microbial biomass, but it is really the
live/active fraction that matters. Since the active fraction vary considerably with environmental
conditions (e.g. soil temperature and moisture explored below), we believe it is reasonable to
expect large coefficients of variation overall in most real-world ecosystems.
**3.2 Environmental Heterogeneity**
So far, we have analyzed in depth the role of variability in microbes and their substrate, but not
in the ecological drivers underlying maximal reaction rates (i.e. $V_{max}$) or half-saturation (i.e. $k_h$).
We start with the observation that both linear first order and non-linear microbial models will
show characteristic scale transitions given heterogeneity in temperature and soil moisture.
Consider the asymptotic convergence of the reverse MM to first order
$$\frac{dC}{dt} = -V_{max}\overline{C} \tag{19}$$

This is mathematically equivalent to the more standard way of writing these models down as
$$\frac{dC}{dt} = -kC \tag{20}$$

In the analysis that follows, we will consider both temperature and soil moisture as factors that
could drive variations in $V_{max}$ over space or time.

**3.2.1 Scale Transition over Temperature Heterogeneity**
To make matters clear, we re-express the rate limiting maximal reaction velocity $V_{max}$ first as a
function of temperature (assuming all else constant):
$$V_{max} = e^{aT} \tag{21}$$

In this case, our integrated flux equation will be:
$$\overline{\frac{dC}{dt}} = \overline{-e^{aT} \times (1 - \epsilon) \times C} \tag{22}$$

Allowing for variability in $T$, this integrated equation will show characteristic scale transitions
given the convex (exponential) relationship with $T$.
Using the Taylor expansion again to second order we have:

$$\overline{V_{max}} \approx e^{a\overline{T}}(1 + \frac{1}{2}a^2 Var(T)) \tag{23}$$

The critical scale transition correction term here is again multiplicative, and we re-express it into
a function of a dimensionless coefficient of variation parameter more suited to ready
interpretation. First, the exponential dependence of respiration on temperature is canonically
codified in terms of $Q_{10}$ scaling. We substitute $a = \frac{log(Q_{10})}{10}$, and end up with:

$$1 + \frac{1}{2}a^2 Var(T) = 1 + \frac{1}{2}(\frac{log(Q_{10})}{10})^2 Var(T) = 1 + \frac{1}{200}(log(Q_{10}))^2(SD(T))^2 =$$

$$1 + \frac{1}{200}(log(Q_{10}))^2(\overline{T}CV(T))^2 \tag{24}$$

For a "typical" $Q_{10}$ of 2.5, and a $\overline{T}$ of 25, we see the multiplicative scale transition correction in
figure 2:

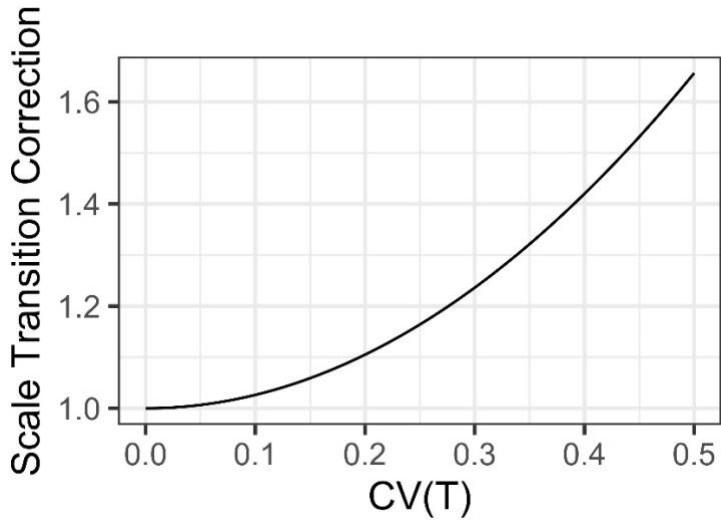


*Figure 2: Scale transition correction for Q10 temperature response scaling given coefficient of*
*variation CV(Temp) from 0 to 0.5*
As is clear in Fig. 2, the scale transition for temperature is extremely convex. Integration of
fluxes over ecosystems with significant heterogeneity in temperature invokes substantial
deviation from a mean-field model. For instance, at a CV of 0.2, the scale transition correction is
1.10, but by a CV of 0.5 it is 1.66. Obviously, the significance of this depends on the scale and
heterogeneity over which an accurate flux model is desired. For a smaller footprint eddy
covariance tower (e.g. Gomez-Casanovas et al., 2018) over a uniform habitat type, soil (and near
surface) temperatures probably do not vary by much more than 20%. Regardless, our general
mathematical analysis quantifies and clarifies exactly how the scale of variation influences the
degree of the scale transition correction.
Notably, the only difference between the scale transition correction for first order and for reverse
Michaelis-Menten kinetics is that in the latter there would be additional correlation terms to
consider, e.g. the correlation between temperature and $V_{max}$, temperature and $k_h$, as well as
temperature and $C$ and $MB$.

### 3.2.2 Scale Transition over Soil Moisture Heterogeneity

Unlike soil temperature, we expect heterotrophic respiration to respond in a unimodal fashion to
soil moisture. At low levels of soil moisture, microbes are moisture limited, and at high levels
they are oxygen limited, with some optimum range of values in the middle. Although a
considerable amount of work has gone into developing soil moisture functions, including both
empirical and theoretical derivations (Yan et al., 2018; Tang and Riley, 2019) , there is no clear
consensus on an optimal representation. Moreover, many of the candidate functions complicate
analysis considerably by virtue of stepwise formulation (Linn and Doran, 1984). Therefore, to
study the implications of the scale transition we proceed via a powerful simplifying abstraction,
and simply represent the soil moisture response as a quadratic of the form:
$$V_{max} = \beta\phi - \beta\phi^2 \tag{25}$$

where $\phi$ represents the soil moisture content. We normalize our function in two senses: first in
output space we assume that it has a maximum of 1 (i.e. represents heterotrophic respiration
relative to a maximum value of 1), and second that the soil moisture content $\phi$ is itself bounded
between 0 and 1, with a peak in the middle at 0.5. Thus, our function captures the unimodal
abstraction in a symmetric form. Given these conditions, there is a unique solution at $\beta = 4$.
We seek the scale transition:
$$\overline{V_{max}} = \overline{4\phi - 4\phi^2} \tag{26}$$

As before, we can approximate this as a mean-field plus a correction to the mean field, which
after some re-writing becomes:
$$\overline{V_{max}} \cong 4(\overline{\phi} - \bar{\phi}^2 - Var(\phi)) \tag{27}$$

We then substitute: $Var(\phi) = \bar{\phi}^2 CV(\phi)^2$, and re-express:
$$\overline{V_{max}} \cong 4(\overline{\phi} - \bar{\phi}^2(1 + CV(\phi)^2)) \tag{28}$$


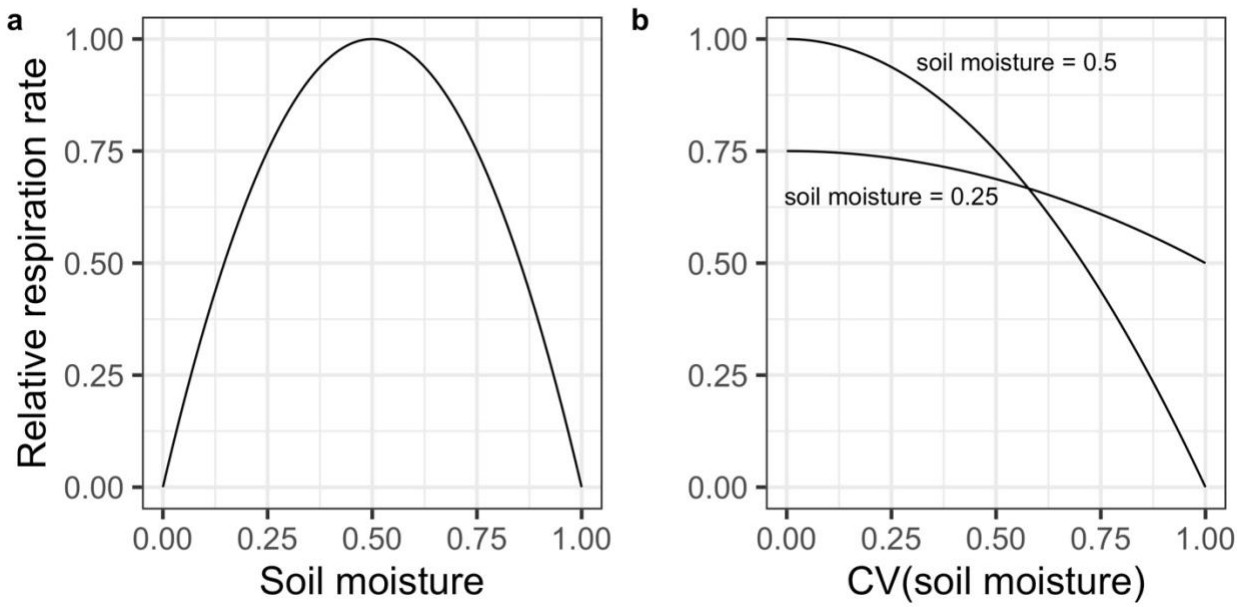

*Figure 3: **a.** heterotrophic respiration as a function of soil moisture given solution in equation 25. Note that soil moisture is normalized with a maximum response at 0.5, where 1 represents complete waterlogging, and respiration is normalized to a maximum of 1. **b.** Scale transition approximate solution for respiration as a function of the dimensionless coefficient of variation of soil moisture where mean field soil moisture is either 0.5 (top curve) or 0.25 (bottom curve, equivalent to 0.75 by symmetry).*

As shown in Fig. 3, where mean field soil moisture is close to the optimal value, scale transition effects are expected to be quite large. For instance, by the time the coefficient of variation is 0.5, efflux would only be 75%, and declines rapidly to 0 as the coefficient of variation approaches 1. Clearly, this latter outcome is not necessarily biologically realistic, and a more detailed numerical experiment should be done to explore scenarios with that much variation. However, our abstractions yield the simple insight that mean-field solutions invariably *overestimate* the real flux, and this overestimation can be considerable. In our experience, soil moisture varies tremendously both over space, especially given contrasts in topography, relief and underlying soils, but perhaps even more so over time, including within a small area, due to day-day and even hour-by-hour variations in precipitation, evapotranspiration and drainage. To the extent that ecosystems deviate from a stable, consistent soil moisture regime, we should expect strong scale transition effects.

Our results on soil moisture relate to the argument by Tang and Riley (2019) that heterotrophic respiration arises from a two-step process whereby substrate must diffuse into the vicinity of microbes, and then be taken up – the latter by a Michaelis-Menten kinetic. However, microscale variations in soil moisture mediate and regulate the first step of the process, so that the "effective substrate affinity" (the $k_h$ term in the Michaelis-Menten model) deviates from the base substrate affinity of the second step. Tang and Riley (2019) point out that the effective substrate affinity therefore reflects microscale heterogeneity, and they argue that experimentalists should account for this when fitting efflux data to models. But what about scaling up in the field from small plots to fields to larger ecosystem units? Fortunately, our analytical framework can be readily queried to account for heterogeneity in substrate affinity ($k_h$).

### 3.2.2.1 Heterogeneity in Substrate Affinity


We proceed by first by holding all terms constant except allowing the half saturation constant $k_h$
to vary, reflecting variations in soil moisture, or frankly any other factor regulating microbial
access to substrate (e.g. soil aggregation, organomineral complexes, etc. Schmidt et al., 2011,
Lehmann et al., 2020). As usual, we seek the scale transition over $\frac{V_{max}MB}{k_h+MB} = f(k_h)$. We can
recover this transition quickly from equation (10) by extracting only the term with ($k_h$), or
rederive from scratch holding everything else as constant. The result is that the dimensionless
scale transition correction term is:
$$1 + \frac{1}{(1+\lambda)^2} CV(k_h)^2 \tag{29}$$

Intriguingly, this result shows that heterogeneity in substrate affinity *per se* results in a convex
correction term, implying that mean-field models will under-estimate rather than overestimate
the resulting fluxes. Given that the correction is proportional to the inverse of the square of $\lambda$,
this correction converges rapidly to 1 (no scale transition) as mean microbial biomass increases
(Figure 4). Nevertheless, where heterogeneity is high and $\lambda$ is around 1 or lower, the correction
could be substantial.
More broadly, our analysis highlights that, under non-linear Michaelis-Menten kinetics for
representing carbon processing, the impact of environmental heterogeneity acting on the
substrate affinity parameter is opposite of when it acts on the $V_{max}$ parameter. Thus, if we
represent soil moisture as a modifier to the $V_{max}$ in the numerator, heterogeneity in soil moisture
should result in lower carbon efflux than mean field, whereas if we represent soil moisture
heterogeneity by way of substrate affinity it is the opposite. At first glance, this finding appears
inconsistent with Tang and Riley (2019). However, we note that their full kinetic formulations
include soil moisture acting in both roles ultimately, and therefore resulting in the familiar
unimodal soil moisture-respiration relationship. For instance, their application of 'Dual Monod'
kinetics include soil moisture driving effective substrate affinity terms for both carbon and water,
as well as an effective fraction of active microbes (which modifies the numerator).
Thus, for the analysis of upscaling fluxes in the presence of soil moisture heterogeneity, we
expect the concave corrections of Figure 3b to hold, regardless of the fine-scale details of the soil
moisture function used.

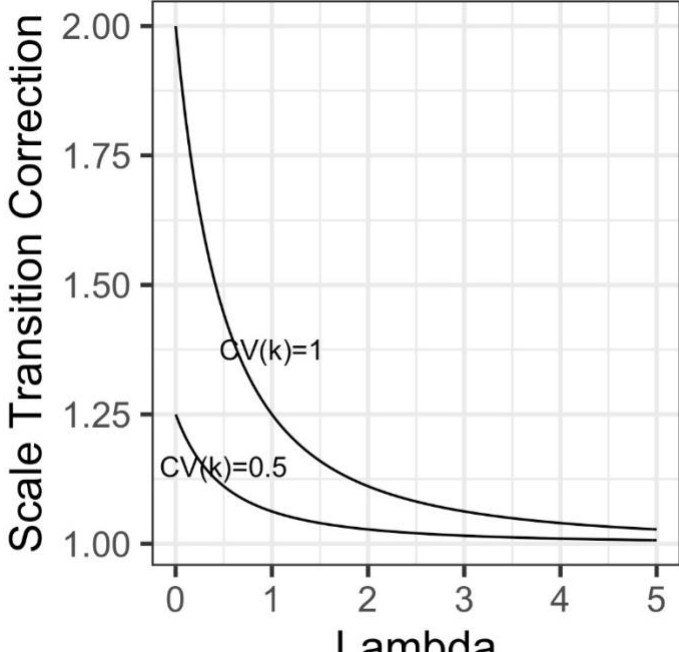


*Figure 4: Scale transition factor for variations in the substrate affinity ("k") parameter in Michaelis-Menten kinetics as a function of dimensionless λ (the ratio of mean-field microbial biomass over mean-field k), for two scenarios of variability in affinity, one where coefficient of variation = 1 (top curve), the other where coefficient of variation = 0.5 (bottom curve).*

### 3.3 Lessons for Scientific Inference

We close our discussion by considering the implications of the scale transition for advancing the state of biogeochemical modeling. Critically, the representation of non-linear (microbial driven) kinetics is a crucial modeling choice with large implications for long-term SOC forecasts. Traditional first-order process-based models dodge explicit representation of these kinetics, but nonetheless have worked well in practice. This state of affairs persists because both non-linear and linear kinetics are capable of representing coarse-scaled biogeochemistry reasonably well, at least in certain respects. Since first order kinetics are known to be a crude approximation, the crucial question for practice is not whether they are "true", but rather whether there is significant, systematic information loss inherent to their use. Fortunately, the scale transition offers a clear, clean path to discriminate between these alternative model formulations.

As noted throughout, the dimensionless term λ plays a critical role in linking the non-linear (Michaelis-Menten) kinetics to the first order kinetics. As λ increases, the non-linear kinetics converge to first order. Thus, in seeking to infer where the non-linear kinetic models provide substantial advantages, ensuring that λ is not too large (>>1) is the first priority.

Previous work Sihi et al. (2016) has approached this question theoretically, from first principles. Here, we point out that demonstrating substantial deviation from mean-field model when fitting

non-linear kinetics to data is both a necessary and sufficient condition for inferring that $\lambda$ is not
too large. Thus, we recommend that time series of flux data be fit to both a first order and a non-
linear kinetic model, where crucial covariates including substrate (SOC), microbial biomass, and
possibly environmental parameters such as temperature, have been measured sufficiently well to
quantify the relevant variances and covariances. Where predictive performance and forecasting
are the primary goals, we recommend careful consideration of model parameterizations
(i.e. based on leave-one-out cross validation), and model combination via "stacking" where it is
difficult to infer a decisive "winner" (Yao et al., 2018) acknowledging that carrying this out is a
significant enterprise.
In addition to the role of $\lambda$, our analysis also cleanly shows the contribution of other terms to the
scale transition, and thus alternative metrics to assess. First and foremost, accounting for the
spatial colocation of microbial biomass and substrate (according to equation 16 above) or the
various correlation terms between microbial biomass and kinetic/environmental factors in
equation 10. Moreover, recent theoretical developments offer quantitative insights into the
interpretation of the half saturation constant (or the substrate affinity parameter) and thus
$\lambda$ (Tang and Riley, 2019). Tang and Riley (2019) decompose microbial access to substrate into a
two-step process, which is often strongly modified by soil moisture. Moreover, conceptual
advances suggest that colocation is a potentially important factor in organic matter
decomposition vs. stabilization (Schimel and Schaeffer 2012, Lehmann et al. 2020). Here, we
show that both affinity and colocation are co-dependent in their effects on scale transition.
In addition to fitting fully parameterized flux models (as above), simpler statistical models could
be fit examining the role of variations in microbial biomass, or colocation of microbial biomass
and SOC, in explaining across-site variations in ecosystem respiratory fluxes ($F$). **A substantial**
**role for either correlation of MB and C, or their variability, would constitute ipso facto**
**evidence of the preferability of well-formulated non-linear kinetic models**. On the other
hand, small roles for colocation, or evidence of large values of $\lambda$ in practice would suggest
minimal advantage to abandoning first order models in favor of more complex microbial models.
A meta-analytical approach across sites will benefit greatly from our formulation in terms of
dimensionless quantities like $\lambda$ and the various coefficients of variation.
We further note that the scale transition presented here is closely related to global sensitivity
analysis (GSA, Saltelli et al. 2010). In its fundamental setup, a GSA tests effects of variability in
parameters. While GSA has been typically used towards characterizing the uncertainty of
parameters, it is directly applicable to spatial and temporal variability. For example, the first
order results of a GSA (or the result of a one at a time parameter substitution), provides the
contribution of that parameter to the scale transition. Similarly, the 'all but one' perturbation
offers insights into how the net effect of all parameters (and variables) violates the mean field
approximation. Therefore, a computationally expensive GSA can be leveraged to garner further
insights on top of sensitivity effects, allowing for the characterization of the scale transition.
Indeed, a computationally intensive approach to simulating scale transitions was utilized by
Chakrawal et al. (2020) to good effect. However, we suggest future computational studies build
off of the dimensionless approach studied here, including those extended to multiple microbial
populations which would result in multiple dimensionless lambdas and corresponding
multiplicative contributions to the scale transition. Obviously, the parameter space needs to be
properly chosen (or subsetted) to reflect appropriate means, variabilities, and perhaps most
challenging - correlations. Equation 10 would then provide analytical, albeit approximate, insight
into the scale transition effects, while the GSA would enable study of any shortcomings from
approximation, and also allow for quantification of individual variable importance for those
parameters that enter into the dynamics in multiple places.
Finally, our analysis of environmental factors including temperature and soil moisture leads to
readily testable predictions. For temperature, the scale transition is convex and thus, ceteris
paribus, variation in soil temperature should lead to greater effluxes than mean field models
would predict. The implications of this for climate-feedback should be studied in greater detail.
For soil moisture, which varies considerably across both space and especially time, our analysis
based on an idealized quadratic representation yields a concave scale transition correction, i.e.
the mean-field soil moisture will over-estimate efflux. Likewise, when represented in both
substrate affinity and multiplicative active microbial biomass fraction terms, as in Tang and
Riley (2019), the scale transition remains concave. However, environmental factors that act only
through substrate affinity would result in a convex correction as in Figure 4. Once again, we
highlight that the nature of scaling corrections, wherever it is possible to be studied empirically,
can provide insight into the most productive representations of our models.
**4. Conclusions**
Here, we have illustrated how the spatial scale transition can be expressed in dimensionless form,
yielding insight into the systematic operation of Jensen's Inequality in upscaling microbial
decomposition kinetics. Our analysis has identified the central role of the dimensionless quantity
$\lambda$ - representing the ratio of mean-field microbial biomass over its half-saturation value - in
governing the extent of the scale transition correction, expressed here in multiplicative form best
facilitating comparison among systems. For somewhat simplified scenarios - such as restricting
to spatial colocation of substrate and microbes - as $\lambda \to \infty$, the mean-field correction goes to 0
and the model converges to first order.
This dual sense of convergence also provides opportunity to empirically test for the presence of
significant non-linear microbial dynamics in upscaled field data: to the extent that upscaled
fluxes deviate from the flux estimated at mean-field conditions, we have *ipso facto* evidence for
the importance of formulating our biogeochemical models with these non-linear terms.
Conversely, where there is close agreement between mean-field and upscaled fluxes, there are
arguably stronger reasons for retaining first-order process model formulations.
In closing, we would like to point out how this mathematical analysis illustrates the challenge of
scaling quite nicely. In the context of non-linear models, for each parameter that is allowed to
vary in space, there is not only a new variance parameter, but a number of new covariance terms
are induced, growing as the factorial of the number of varying parameters $\binom{5}{2}$! (Fig. 3). Thus, in
the case of the 5 parameter function considered here, the full approximation has 5 mean field
terms, 5 coefficients of variation, 10 correlation coefficients, and the dimensionless quantity $\lambda$.

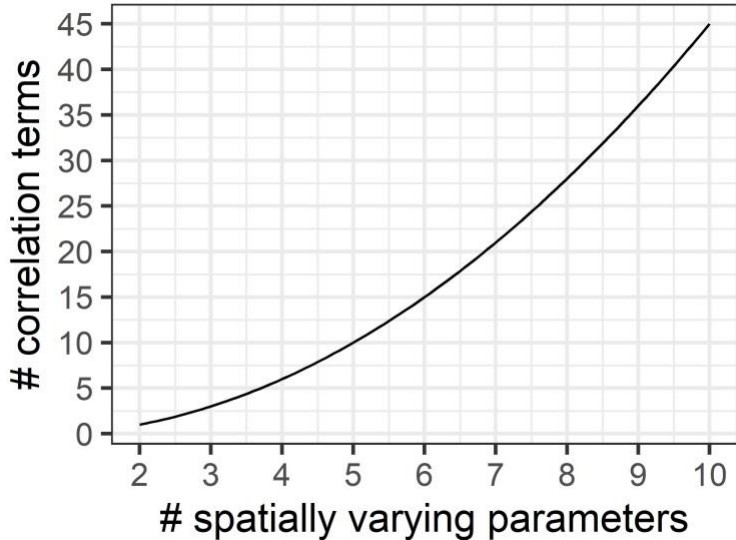


*Figure 3: Model complexity grows exponentially with number of spatially varying parameters. We argue to keep models as simple as possible for both analytical and computational tractability.*

Even with a maximally generic and simplified expression, fitting such non-linear time series models to field data still represents quite a challenge, especially while adequately accounting for and propagating uncertainty. Modelers and theoreticians should appreciate the complexity of the task at hand. Fortunately, our analysis has identified a potentially robust route to limiting model complexity: screen systematically for the importance of various correlations in explaining variations in fluxes. Accordingly, we recommend that research focus first upon spatial colocation of MB and C, which is readily measured, and then to thoughtfully and carefully expand models with additional terms as needed.

**Author Contributions:** CHW conceived the original concept, developed the mathematical analysis, and wrote the manuscript. SG developed the concepts, contributed to the mathematical analysis, and co-authored and edited the manuscript.

**Competing Interests:** None declared.

**Acknowledgements:** We stand on the shoulders of giants: Peter Chesson's research program on Scale Transition was enormously influential. We thank T. Trevor Caughlin for introducing us to Chesson's papers many years ago, and to everyone who has humored discussions of Jensen's Inequality ever since. Will Wieder and an anonymous reviewer provided constructive reviews that improved our manuscript, and Kathe Todd-Brown provided valuable discussion as we worked through revisions.

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
