# Peer review of "Theoretical insights from upscaling Michaelis-Menten microbial dynamics in biogeochemical models: a dimensionless approach"

_Biogeosciences, 2021_

## Author Response (AR1)

Dear Editor Subke,

We have prepared a fully revised version of our manuscript in response to your recommendation of minor revisions and the reviewers' comments. Our revisions are noted with track changes and with highlighting throughout the manuscript.

Overall, we feel the manuscript has been noticeably strengthened as a result of this review process. We developed and incorporated a new analysis of soil moisture responses. The soil moisture discussion is split into two new sections, one addressing how soil moisture affects scale transition via the affinity parameter and one regarding the empirical formulation of soil moisture impact. This addresses both reviewers primary comments regarding the 2-step process typically involved in carbon processing, and the environmental parameterization of soil moisture function. Other suggestions and edits have also been incorporated, and our overall responses are detailed line-by-line below in italics. We hope you find the revised version suitable to be published. Please let us know if any additional changes are needed!

Sincerely,
Chris H. Wilson and Stefan Gerber, University of Florida
* * *
**RC1:** In this study, the authors proposed a dimension-less reformulation of the scale transition method to analyze the effect of spatial heterogeneity on the mean-field description of SOC dynamics by the Michaelis-Menten kinetics. They showed that the ratio between mean microbial biomass and mean half-saturation parameter of the Michaelis-Menten kinetics can serve as the characteristic parameter to measure the strength of spatial heterogeneity and the non-linearity of the SOC dynamics of interest. They argue that such a result can help both theoreticians and empiricists to better interpret observed soil C decomposition dynamics, specifically respiration in their case.

 Overall, I think this is a well-attempted study, and is clearly reported. I have some moderate to major suggestions for the authors to further improve the manuscript.

*CHW: We thank the reviewer for taking the time to read our manuscript carefully and providing us with a constructively critical overall assessment.*

Idea-wise, the paper is very similar to the study by Chakrawal et al. (2020), except that the presentation here is simplified by introducing a characteristic parameter and normalization of the variances by the mean field approximation (making the variance corrections dimension-less in the analysis). However, I think the authors ignored the fact that some of the spatial heterogeneity

can be considered in the nonlinear kinetics by recognizing that decomposition is at least a two-step process: (1) microbes approach the substrate (or vice versa), and (2) microbes assimilate the substrate. Consequently, the nonlinearity with the Michaelis-Menten kinetics emerge from the combination of two linear steps, and one thus should not be surprised to see that the mean-field Michaelis-Menten kinetics cannot upscale robustly. As shown in Tang and Riley (2019), conceptualizing microbial substrate uptake as a two-step processes enables the half saturation parameter to incorporate the spatial heterogeneity to some extent. I thus recommend the authors to clarify this.

*CHW: The reviewer highlights an interesting point here vis a vis the derivation of the Michaelis-Menten kinetics. We agree that more detailed representations of decomposition can have utility in extending our biogeochemical models to smaller scales. The derivation present in Tang and Riley (2019) illustrates how microscale/pore-scale models can be developed that represent how soil moisture mediates diffusional transport of substrate to microbial cells. However, Tang and Riley (2019) nonetheless represent the final uptake of that substrate as following Michaelis-Menten kinetics, which is non-linear. Put together, these processes define what they call "effective substrate affinity" (let's call this k_eff). All in all, k_eff is still a component in a non-linear dynamic and would be expected to vary substantially across space in any field setting. So, we therefore think that our analysis is especially pertinent for the scale transition at this level, and thus complements the microscale analysis of Tang and Riley (2019). In our view, Tang and Riley (2019) provide a very interesting mechanistic basis for incorporating variations in soil moisture into the substrate affinity portion of the non-linear kinetics. Accordingly, we now have two new sections in our paper: 3.2.2 where we derive a new idealized analysis of soil moisture driven scale transitions, and 3.2.2.1 where we specifically investigate the role of heterogeneity in substrate affinity. These sections have also resulted in new figures 3 and 4. We link our analysis of these sections explicitly back to Tang and Riley (2019) in those new sections and through expanded discussion in lines 428-434.*

Additionally, in the "lessons for scientific inference", I would suggest the authors trying to discuss the relationship between their analysis and the parametric sensitivity analysis or uncertainty analysis that are very popular in the modeling community. Apparently, what the authors presented here and also in Chakrawal et al. (2020) are closely related to approaches like global sensitivity analysis that is used to understand the influence of parametric uncertainty on model performance. Has the authors here just rediscovered the global sensitivity analysis in a new context? And what can sensitivity or uncertainty analysis learn from the authors' study?

*CHW: We agree this is a helpful way to strengthen our discussion. In brief, our analysis would be an ideal starting point for parameterizing a global sensitivity analysis (GSA), since it is framed in terms of dimensionless quantities and multiplicative effects which therefore do not depend on the choice of unit scales etc. Although our focus was on highlighting the qualitative insights possible with this setup, we do think that it would make sense particularly for intercomparison among systems with varying soils, hydrology, climate, plant and microbial communities where parameter values would be expected to vary considerably. With a more generic setup such as ours, it is easier to isolate which combinations of parameter values are likely to lead to significant scale transitions. In lines 444-460, we make the connections clear as we see them.*

If properly made, I believe the paper will become much more interesting to general readers. Particularly, I don't see an easy way to apply the authors approach to a model that is presented as numerical code, which technically could involve tens if not hundreds of equations, e.g. for a decomposition model that involves tens of microbial populations that I am work with, I don't see how I can apply the authors' method.

*CHW: We believe that this is still a possibility and should actually also be done in more complex models. It is possible, even for a complex model to calculate the Hessian Matrix (for example from a sensitivity analysis), This may further help in evaluating and identifying the most crucial parameter and identify the variance/covariance structure in an efficient way, given the variability of the system. This will further help with the ultimate goal of integrating structures of co-location into models. We aim to expand this in our revision, and fits well with your prompt regarding the GSA.*

Finally, how should one analyze the temporal heterogeneity blended with the spatial heterogeneity using the authors method?

*CHW: This is an excellent question. Although we believe it is beyond the scope of this paper, our approach offers a template. The temporal tendency of the mean field can be expanded to include the scale transition terms arising from the variance-covariance matrix and the Hessian, ideally in dimensionless form as in the present contribution. The components arising from the hessian matrix can be evaluated at any time. However, the coefficients of variation that result are much more difficult and may depend to a great degree on the specific problem. Ultimately, the details of how microbial dynamics develop over time and space in response to resources and environmental conditions that vary both predictably and unpredictably is not analytically tractable, and therefore no closed-form, time dependent function for the relevant covariances/coefficients of variation can be written down. In the extreme, responses to catastrophic disturbances lead to deep and transient alteration of the covariance structure for which there may not be a readily available solution. However, for smaller time horizons, absent catastrophic weather or geological events, we believe that empirical sampling of the correlations and variabilities can be used to gain insight into scale transition effects influencing observed gas exchanges.*

**Reference**

Chakrawal, A., Herrmann, A.M., Koestel, J., Jarsjo, J., Nunan, N., Katterer, T., Manzoni, S., 2020. Dynamic upscaling of decomposition kinetics for carbon cycling models. Geoscientific Model Development 13, 1399-1429.

Tang, J.Y., Riley, W.J., 2019. A theory of effective microbial substrate affinity parameters in variably saturated soils and an example application to aerobic soil heterotrophic respiration. Journal of Geophysical Research-Biogeosciences 124, 918-940.

**Citation**: https://doi.org/10.5194/bg-2021-108-RC1

**RC2:**

**Summary**

Wilson and Gerber present a thoughtful analysis on challenges related to scaling microbial dynamics to ecosystem scales. Their work takes a deep dive into the mathematics of these transition across scales. The work is well reasoned, well supported, and well written. My chief suggestion with this paper is to encourage the authors to take a step back from mathematical rigor of their analysis to connect their ideas more broadly with theories and measurements related to SOM turnover, persistence, and vulnerability. My major suggestions are aimed at making these suggestions

*CHW: We appreciate the thoughtful review from Will Wieder, and feel that the changes made in response to the review have substantially improved the manuscript.*

**Major Suggestions**

The mathematical focus on variability (especially related to substrates and microbes) that this paper explores seems connected to the more theoretical ideas in Schimel and Schaffer 2012 and Lehman et al 2020. I wonder if revisions can reach a broader audience by connecting the quantitative depth of this paper with these broader concepts?

*CHW: Throughout manuscript we have expanded our references to the literature suggested here and elsewhere by Will Wieder. For instance, Schimel and Schaffer (2012) and Lehman et al. (2020) are a component of our strengthened discussion in lines 428-434. We also work new references into our new sections on soil moisture and introduction (e.g. line 79-84, 318, 367-368). Our analysis does not address molecular diversity as in Lehman et al. (2020), whereas we address spatial heterogeneity through the correlation between microbes and substrate as well as their respective variabilities. Interestingly, the empirical soil moisture function may be considered as a mean field characterization of access as discussed in response to referee 1, and further elaborated below.*

Water seems like the big unknown here. If the colocation of microbes and substrates is largely dependent on liquid water availability (as well as SOM-mineral interaction) then high heterogeneity of soil moisture within sites seriously complicates the feasibility of actually capturing the local scale heterogeneity for which the authors seem to be advocating. This is not a deal-breaker for publication, but it seems like a topic that could be discussed? More details in the minor comments below.

*CHW: We agree that soil moisture is a critical component of environmental variability, and very likely to induce scale transition effects. As discussed above, water influences access, but also oxygen limitation. In the framework of Tang and Riley (2019), the formulation of an appropriate soil moisture function can be thought of as defining an "effective substrate affinity" term, which in itself accounts for some of the microscale heterogeneity. But we absolutely agree, as noted above, that over larger spatial scales, soil moisture variation should be studied within the analytical framework of the scale transition. One challenge is that many soil moisture functions*

*(e.g. Yan et al. 2018) are piecewise, rather than smoothly continuous, rendering their analysis substantially more complicated. In the face of these complications, we can see a few strategies. First, the soil moisture function could be replaced by a simplified, perhaps polynomial or Gaussian approximation, whose analytical properties would be amenable to our treatment. Alternately, given at least a continuously differentiable function and a probability kernel, a full convolution integral could be set up and either solved if possible or studied numerically. For this revision we used a polynomial (quadratic) approximation and derived new analytical results and figures linking dimensionless variability to scale transition corrections.*

*Our new results are detailed in section 3.2.2 (lines 313-364) where we provide the dimensionless analysis of a quadratic soil moisture response, and in section 3.2.2.1 (lines 365-398) where we analyze the role of substrate affinity heterogeneity per se. We find the possibility of potentially large scale transitions, and we note that these are especially likely with soil moisture in a temporal sense, given how much it fluctuates with rainfall, evapotranspiration, and drainage.*

Line 230, this statement seems values for results in 1c, which converges at 0.9. This seem relevant, especially if one take home message from the text as presented is that 'a first order model is good enough'. This may be true, but what are the implication for having a scale correction factor that does not equal 1, even at large lambdas? Are the conditions required for this to occur plausible in natural systems?

*CHW: This is a sharp observation. However, for large enough lambda, the convergence does proceed to 1 as indeed it must according to equation 15. Thus, the appearance of convergence to 0.9 is an artefact of the axis range. However, we believe that this phenomenon is actually useful to highlight how \*slow\* the convergence is for large variabilities. We have revised our legend to make this explicit and clear! (lines 236-238). As to the realism, we believe that a useful next project would involve a meta-analytic collaboration to compile empirical data on the relevant correlation and variability terms where they can be gleaned from published research. Lines 414-424 and 499-506 discuss.*

**Minor and Technical Comments**

Throughout, I'd encourage the authors to be precise and consistent in their terminology for things like "scale transition correction", 'scale correction factor', 'mean field correction', etc. This will avoid confusion and help clarify the ideas in the text.

*CHW: We will do that in the revision – good suggestion. Line 263 has eliminated 'mean field correction' and replaced with 'scale transition correction'. Lines 234, 298, 302 have been harmonized to 'scale transition correction'.*

Line 54, I might add Wang et al. 2016 to this list.

*CHW: added line 54.*

Line 78, can the author's just write out process-based models throughout? There are enough acronyms in the text, removing this non-standard one that's sparingly used will aid

readability.Line 80,  Although not related to trace gas fluxes, Bradford et al 2017, 2021 raise similar concerns related to litter decomposition rates.  These reference are also good ones related to the major concern about soil moisture, above.

*CHW: Done. We have replaced PBM everywhere with process-based model instead, and added Bradford reference to line 78.*

Line 89,  remove 'enormous'

*CHW: Done.*

Line 90 remove 'universal'

*CHW: Now on line 97, we say 'new' instead.*

Line 111 remove this duplicated equation, or number it?

*CHW: This is now line 120 and equation # 4.*

Line 175, and potentially in the introduction (see Buchkowski et al 2017).

*CHW: We have added Buchkowski to this reference list and to the introduction (line 185-186, and 75-77 respectively).*

Line 210.  This does seem like valuable insight.  Notably, the variation in soil moisture (which I'd argue impacts substrate availability) is very high (Loescher et al. 2014).  I wonder what implications this has for looking at flux estimates within and among sites?

*CHW: As noted in responses above and further below, we agree that soil moisture is a really important component of both spatial and temporal variability, and we believe our new sections (3.2.2, 3.2.2.1) and strengthened discussion on this topic have improved the manuscript.*

Should the y-axes on fig 1 be held constant so that it's more obvious that at the CV(MB) increases the magnitude of the scale factor changes? Also, should Fig 1  & 2 have the same y-axis label, they're both showing a Scale correction factor.

*CHW: We have adopted Will Wieder's suggestions to fix the axis ranges and thus highlight the large differences in scale transition. Moreover, we have harmonized y-axis labels in both Figures 1,2, and 4 to "Scale Transition Correction" to be consistent with our text harmonization, as suggested.*

Line 235 maybe replace 'virtus' with something link 'benefit'

*CHW: Done (line 242).*

Line 255, microbial biomass is commonly used in models, but really it's the 'active' microbial biomass that matters here, which seems to be much more variable (and harder to quantify).

*CHW: This is a good point. We clarify this in lines 266-270 where we write, "Finally, we note that throughout, our development of these kinetics assume proportionality to microbial biomass, but it is really the live/active fraction that matters. Since the active fraction vary considerably with environmental conditions (e.g. soil temperature and moisture explored below), we believe it is reasonable to expect large coefficients of variation overall in most real-world ecosystems."*

Section 3.2. This is a nice example for temperature, but given much higher variance in moisture, (again see Loescher et al 2014) it seems like the real environmental variance we need to care about within sites is moisture. Maybe this doesn't need a mathematical proof, but a brief discussion.

*CHW: See our response regarding soil moisture to earlier comments. We think addition of this mathematical treatment has strengthened paper.*

Line 321, this sounds ideal, but I wonder where this is ever going to occur for all the variables of relevance, again see Loescher et al (2014)?!

*CHW: We have amended the sentence with the reference, and acknowledged that this is a tall order (427-428).*

Extending a bit on the comment above I have additional thoughts, listed below. These are NOT intended to be stinging critiques of the work presented, but I would encourage the authors to discuss some of the more practical challenges that would be involved with putting the research plan they outline into place:

- SOC does not equal available C and microbial biomass does not equal active biomass.

*CHW: Agreed, and we have clarified for instance the issue with live microbial biomass.*

- How do these measurements integrate with depth?

*CHW: In the context of looking at respiration flux and NEE, spatial heterogeneity also includes depth heterogeneity. Perhaps a first pass would be to integrate each soil horizon (or depth interval within a decomposition model) separately, where each layer will have their own terms. All in all, this strengthens our argument that a flux-tower derived parameter is almost surely not a true physiological parameter, but carries the issue of scale transition.*

- How do we bridge the jump between the variability in soil and trying to infer heterotrophic respiration fluxes from NEE measured in the flux tower.

*CHW: Ultimately, there are many considerations here. Our preference would be to develop a large joint generative statistical model that assimilates all the available data, estimates parameters, and can then be used for forecasting and inference via the posterior predictive distribution (e.g. Caughlin et al. 2021, Dietze 2017). Based on our analysis in this paper, we*

*believe that an important component of this model development will be to include relevant scale transition terms as needed in both measurement and process specifications in this model. For instance, rather than fitting a mean-field model for the flux (F) as in equation 3, we might include say the spatial colocation terms and the corrected model in equation 15 (were the right data available). Lines 414-424 and 499-506 discuss.*

- Even if all this could be measured with enough fidelity at a single site how do we extend such insights to resolution of a grid cell in an ESM (nominally 1 degree or 100x100 km).

*CHW: We agree that this is a tall order, but having an awareness of scale transition, and perhaps an understanding where scale transition are likely to be especially large, is critically important. Ultimately, our hope is that this work is the start of a conversation about systematic scale transition effects, and that reconciliation between theory and data via rigorous statistical models will help untangle critical aspects to constrain and model when going from plot to field to site to ESM grid cell.*

- The logic and math presented here is fascinating, and it does help prioritize the measurements that need to be taken, but I do wonder if it's realistic to make the measurements given existing technology & infrastructure?

*CHW: We agree that in some cases the measurements may be logistically unfeasible. The first step, in our view, is to work within existing footprints of eddy covariance towers, and fit microbial models with and without the scale transition terms, which in turn would be informed with systematic sampling of the relevant covariates. Measuring spatial variations in SOC, extractable microbial biomass (or alternately, SIR estimates of active biomass), soil moisture and texture, and so on should be tractable at this scale. But, overall, we agree with Will Wieder that this is a challenging task that will require collaboration between theorists, statistical modelers, and empiricist!*

Line 335.  I'd push back a bit on this statement, because if the aim of these models is to faithfully capture soil flux measurements, then I'd assume there's not much benefit in using anything but a first order model.  If, however, the aim of these models is to more broadly explore our theoretical understanding of microbial and soil controls over SOM persistence and vulnerabilities, then microbial explicit models may be useful (see Wieder et al. 2015).

*CHW: We agree with the notion that it must be a goal to work with conceptually sound models. But if parameterization is a goal, it follows that it will be difficult for more complex and perhaps physiologically more realistic model to obtain proper parameterization. Conceptually sound models are still useful to define range and limits of simplified, first order models. Moreover, even first order models may have critical scale transitions in their response to environmental drivers such as temperature and soil moisture!*

 I don't love presenting a new figure in the conclusion of a paper, but this is more of a stylistic comment than a serious critique of the work.

*CHW: We think the nature of this figure is conceptual and illustrates an issue following the discussion, and therefore deserves the exception.*

**References:**

Buchkowski, R. W., Bradford, M. A., Grandy, A. S., Schmitz, O. J., & Wieder, W. R. (2017). Applying population and community ecology theory to advance understanding of belowground biogeochemistry. *Ecology Letters, 20*(2), 231-245. doi: 10.1111/ele.12712.

Bradford, M. A., Veen, G. F. C., Bonis, A., Bradford, E. M., Classen, A. T., Cornelissen, J. H. C., Crowther, T. W., De Long, J. R., Freschet, G. T., Kardol, P., Manrubia-Freixa, M., Maynard, D. S., Newman, G. S., Logtestijn, R. S. P., Viketoft, M., Wardle, D. A., Wieder, W. R., Wood, S. A., & van der Putten, W. H. (2017). A test of the hierarchical model of litter decomposition. *Nat Ecol Evol, 1*(12), 1836-1845. doi: 10.1038/s41559-017-0367-4.

Bradford, M. A., Wood, S. A., Addicott, E. T., Fenichel, E. P., Fields, N., González-Rivero, J., Jevon, F. V., Maynard, D. S., Oldfield, E. E., Polussa, A., Ward, E. B., & Wieder, W. R. (2021). Quantifying microbial control of soil organic matter dynamics at macrosystem scales. *Biogeochemistry*. doi: 10.1007/s10533-021-00789-5.

Lehmann, J., Hansel, C. M., Kaiser, C., Kleber, M., Maher, K., Manzoni, S., Nunan, N., Reichstein, M., Schimel, J. P., Torn, M. S., Wieder, W. R., & Kögel-Knabner, I. (2020). Persistence of soil organic carbon caused by functional complexity. *Nature Geoscience, 13*(8), 529-534. doi: 10.1038/s41561-020-0612-3.

Loescher, H., Ayres, E., Duffy, P., Luo, H., & Brunke, M. (2014). Spatial Variation in Soil Properties among North American Ecosystems and Guidelines for Sampling Designs. *Plos One, 9*(1), e83216. doi: 10.1371/journal.pone.0083216.

Schimel, J. P., & Schaeffer, S. M. (2012). Microbial control over carbon cycling in soil. *Front Microbiol, 3*, 348. doi: 10.3389/fmicb.2012.00348.

Wang, Y. P., Chen, B. C., Wieder, W. R., Leite, M., Medlyn, B. E., Rasmussen, M., Smith, M. J., Agusto, F. B., Hoffman, F., & Luo, Y. Q. (2014). Oscillatory behavior of two nonlinear microbial models of soil carbon decomposition. *Biogeosciences, 11*(7), 1817-1831. doi: 10.5194/bg-11-1817-2014.

Wieder, W. R., Allison, S. D., Davidson, E. A., Georgiou, K., Hararuk, O., He, Y., Hopkins, F., Luo, Y., Smith, M. J., Sulman, B., Todd-Brown, K., Wang, Y.-P., Xia, J., & Xu, X. (2015). Explicitly representing soil microbial processes in Earth system models. *Global Biogeochemical Cycles, 29*(10), 1782-1800. doi: 10.1002/2015gb005188.

**Citation**: https://doi.org/10.5194/bg-2021-108-RC2

Additional References for reply:

Caughlin, T. T., Barber, C., Asner, G. P., Glenn, N. F., Bohlman, S. A., & Wilson, C. H. (2021). Monitoring tropical forest succession at landscape scales despite uncertainty in Landsat time series. *Ecological Applications*, *31*(1), e02208.

Dietze, M. C. (2017). Prediction in ecology: A first-principles framework. *Ecological Applications*, *27*(7), 2048-2060.

Yan, Z., Bond-Lamberty, B., Todd-Brown, K. E., Bailey, V. L., Li, S., Liu, C., & Liu, C. (2018). A moisture function of soil heterotrophic respiration that incorporates microscale processes. *Nature communications*, *9*(1), 1-10.